

# Emulating twisted double bilayer graphene with a multiorbital optical lattice

**Junhyun Lee and J. H. Pixley**

Department of Physics and Astronomy, Center for Materials Theory,
Rutgers University, Piscataway, NJ 08854, USA

## Abstract

This work theoretically explores how to emulate twisted double bilayer graphene with ultracold atoms in multiorbital optical lattices. In particular, the quadratic band touching of Bernal stacked bilayer graphene is emulated using a square optical lattice with $p_x$, $p_y$, and $d_{x^2-y^2}$ orbitals on each site, while the effects of a twist are captured through the application of an incommensurate potential. The quadratic band touching is stable until the system undergoes an Anderson like delocalization transition in momentum space, which occurs concomitantly with a strongly renormalized single particle spectrum inducing flat bands, which is a generalization of the magic-angle condition realized in Dirac semimetals. The band structure is described perturbatively in the quasiperiodic potential strength, which captures miniband formation and the existence of magic-angles that qualitatively agrees with the exact numerical results in the appropriate regime. We identify several magic-angle conditions that can either have part or all of the quadratic band touching point become flat. In each case, these are accompanied by a diverging density of states and the delocalization of plane wave eigenstates. It is discussed how these transitions and phases can be observed in ultracold atom experiments.


doi:10.21468/SciPostPhys.13.2.033

# 1 Introduction

Emulating quantum many-body Hamiltonians using ultracold gases of atoms in an optical lattice have undergone significant advances in recent years [1,2]. The ability to realize strongly correlated Hubbard models has been achieved [3,4] as well as the ability to program disordered or quasiperiodic potentials into the system to induce localization phenomena [5,6]. On the other hand, recent developments in the ability to accurately twist van der Waals heterostructures [7–11] have opened the door for a new level of control over two-dimensional solid-state materials. Recent theoretical work has proposed realizations of this phenomena in ultracold atomic systems by either twisting the optical lattice [12] or its spin state [13], as well as emulating the effects of a twist using incommensurate, quasiperiodic potentials [14–18]. Recently, experiments have successfully twisted optical lattices holding a Bose-Einstein condensate opening the door for experimental realizations of twistronics of ultracold atoms [19].

A fascinating aspect of twisted van der Waals heterostructures is that despite the underlying materials being weakly correlated, twisting induces (an almost periodic) moiré pattern on a much larger superlattice length scale that strongly renormalizes the electronic dispersion inducing isolated flat bands that quench the kinetic energy and promote strong correlations [20–24]. This approach has been remarkably successful as there are now experimental discoveries of correlated insulators and superconductors in twisted bilayer graphene [7–11], twisted double bilayer graphene [25–27], twisted tri-layer graphene [28–31], and in twisted transition metal dichalcogenides [32,33]. Moreover, topological states have also been observed with a quantized anomalous Hall effect when magic-angle graphene is aligned with the bornon-nitide substrate [10,34,35].

As is now becoming clear, twisting represents a common approach to downfold and reconstruct the underlying band structure that now lives in a mini Brilloiun zone due to a much larger approximate moiré unit cell in real space (e.g. see Fig. 1). While originally twisting was proposed to manipulate the low-energy massless Dirac excitations in graphene it is now understood that it can also have dramatic effects on higher order nodal points as well as states with a Fermi surface [32,33,37,38]. In particular, the quadratic and cubic band touchings that occur in AB Bernal stacked bilayer [25–27] and ABC stacked trilayer graphene [39] respectively have both been manipulated via a twist to induce correlated insulators and superconductors. While at face value these nodal touching points appear similar, in two-dimensions however, any touching point with an integer power that is larger then linear will have a finite density of states at the Fermi energy and hence be metallic, which is in stark contrast to the exact zero density of states in a Dirac semimetal. As a result, it is unclear what aspects of twisting a Dirac semimetal, such as a magic-angle with a vanishing velocity that coincides with the development of a finite density of states and the existence of flat isolated bands can carry over to

twisting higher order nodal touching points. For example, in twisted double bilayer graphene, a magic-angle condition where the quadratic band touching point becomes flat only persists in the absence of trigonal warping terms and particle hole asymmetric perturbations [40]. In light of the wide variety of twisted van der Waals heterostructures it is an interesting problem to understand how to emulate other classes of twisted band structures.

In this manuscript, we build on this perspective to emulate twisting quadratic-band-touching (QBT) bands as in double bilayer graphene (i.e. twisting two different bilayers of AB-Bernal stacked bilayer graphene) in ultracold atoms. Our proposal utilizes multiorbital optical lattices that have been realized in Refs. [41–43], depicted in Fig. 1. In particular, we consider a three-orbital model on the square lattice introduced in Ref. [36] that has a QBT in its dispersion relation. The effect of twisting is emulated via a two-dimensional quasiperiodic potential, which can be realized using recently developed techniques that have observed two-dimensional localization transitions [44,45]. We show that the general notion of a magic-angle condition, where the Dirac cone velocity vanishes in the presence of an incommensurate tunneling or potential, naturally generalizes to the case of a quadratic band touching. Here, the quadratic band touching affords a lot more flexibility then a Dirac point allowing for magic-angles where only part of the quadratic band touching point becomes flat in addition to fully flat nodal points. It is demonstrated that in the incommensurate limit each magic-angle condition becomes an eigenstate phase transition, where the plane wave eigenstates Anderson delocalize in momentum space. As a result, the system transitions into a metallic phase with a diverging density of states. In the vicinity of the quadratic band touching point we find the incommensurate potential drives the formation of a sequence of minibands that live on the moiré superlattice. Last, we discuss how each phase and phase transition we have found can be probed in experiments on ultracold Fermi gases.

The remainder of the manuscript is organized as follows: In Sec. 2 we define the model and parameter regime we consider. We also define key observables such as the effective mass of the QBT band and inverse participation ratio, and introduce the numerical approaches. In Sec. 3 we investigate how the excitation spectrum is affected by the quasiperiodic potential, first calculated by perturbation theory and next with finite-size numerics. We see how the dispersion is renormalized, especially how the band flattens and the minibands emerge. We study the eigenstate properties of the band flattenings in Sec. 4 and how it relates to the Anderson-like localization transition. We discuss the experimental realization and noteworthy outlooks in Sec. 5 and conclude in Sec. 6.

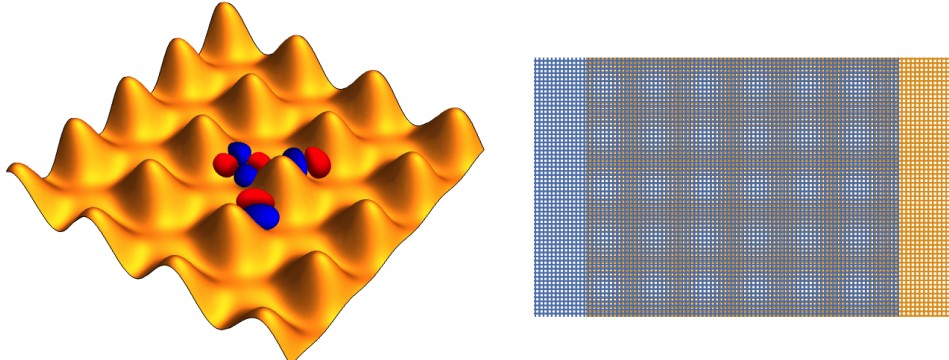

Figure 1: **Multi-orbital optical lattice and moiré pattern:** Schematic description of the optical lattice from Ref. [36]. The $p_x$, $p_y$, and $d_{x^2-y^2}$ orbitals on each site are depicted in the middle three sites, where the different color indicates the sign of the wavefunctions. The right figure shows a top view of two incommensurate square lattice, demonstrating the moiré lattice that arises due to their interference pattern.

## 2  Model and Approach

To emulate twisted double bilayer graphene we take a Hamiltonian that is given by

$$H = H_0 + H_V,\tag{1}$$

where $H_0$ is the dispersion that must encode a quadratic band touching at an isolated point in the Brillouin zone, and $H_V$ emulates the effect of a twist through an incommensurate quasiperiodic potential. To construct $H_0$ we consider a three-band model from Ref. [36] on the square lattice, representative of the orbitals $p_x$, $p_y$, and $d_{x^2-y^2}$ at each site $\mathbf{r}$ of an optical lattice, see Fig. 1. The details on the realization of the model, including the experimental parameters for the optical lattice and pre-tight binding approximation band structures, are presented in Appendix A. In the following we focus on the tight binding limit that is given by

$$H_0 = \sum_{\mathbf{k}} \Psi_{\mathbf{k}}^{\dagger} \mathcal{H}_0(\mathbf{k}) \Psi_{\mathbf{k}},\tag{2}$$

where $\Psi_{\mathbf{k}}^T = (d(\mathbf{k}), p_x(\mathbf{k}), p_y(\mathbf{k}))$ and

$$\mathcal{H}_0(\mathbf{k}) = \begin{pmatrix} -2t_{dd}(\cos k_x + \cos k_y) + \delta & 2it_{pd}\sin k_x & 2it_{pd}\sin k_y \\ -2it_{pd}\sin k_x & 2t_{pp}\cos k_x - 2t'_{pp}\cos k_y & 0 \\ -2it_{pd}\sin k_y & 0 & 2t_{pp}\cos k_y - 2t'_{pp}\cos k_x \end{pmatrix}.\tag{3}$$

Here, $t_{\alpha\beta}$ is the hopping parameter between $\alpha$ and $\beta$ orbitals, where $t_{pp}$ denotes the $p_{x/y}$-orbital hopping in $x/y$ direction while $t'_{pp}$ is the $p_{x/y}$-orbital hopping in $y/x$ direction. $\delta$ is the relative chemical potential of the $d_{x^2-y^2}$ orbital to the $p$ orbitals, which controls the hybridization between the $d$ and $p$ orbitals. To start with a clean quadratically touching single particle spectrum with no other energy levels in the vicinity of the touching energy, in this paper we concentrate on the strong hybridization limit ($0 < \delta < 4t_{dd} + 2t_{pp} - 2t'_{pp}$). For the detailed tight binding model constructed via an optical lattice and its weak hybridization limit, see Ref. [36].

The three band model in Eq. (3) generally has degeneracies at the $\Gamma$ and the $M$ points, and in the strong hybridization limit only one band connects the two degeneracies in the $\Gamma M$ line as shown in Fig. 2. Both degenerate points disperse quadratically, and we call these QBT points. We choose the parameters such that the quadratic dispersion is isotropic ($t_{dd} = t_{pp} = 3t'_{pp} = \delta \equiv t$, and $t_{pd} = \sqrt{(t_{pp} - t'_{pp})(2t_{pp} - 2t'_{pp} + 4t_{dd} + \delta)/2}$), however our discussion is not specific to this fine tuning of parameters. For the following discussion, we focus on the QBT with the lower energy (with energy $E_{\mathrm{QBT}}$) located at the $M$ point as this isolated with no other "parasitic" bands crossing at this energy.

To characterize the properties of the QBT, we expand $\mathcal{H}_0(\mathbf{k})$ around the $M$ point up to quadratic order in $\mathbf{q} \equiv \mathbf{k} - (\pi, \pi)$:

$$\mathcal{H}^{(2)}(\mathbf{q}) = \begin{pmatrix} t_{dd}(4 - q_x^2 - q_y^2) + \delta & -2it_{pd}q_x & -2it_{pd}q_y \\ 2it_{pd}q_x & t_{pp}(-2 + q_x^2) + t'_{pp}(2 - q_y^2) & 0 \\ 2it_{pd}q_y & 0 & t_{pp}(-2 + q_y^2) + t'_{pp}(2 - q_x^2) \end{pmatrix}.\tag{4}$$

The eigenenergies of $\mathcal{H}^{(2)}(0)$ are $4t_{dd} + \delta$ and doubly degenerate $-2t_{pp} + 2t'_{pp}$'s, where the latter is the value of the energy of the QBT, namely $E_{\mathrm{QBT}} = -2t_{pp} + 2t'_{pp}$. The QBT can be further characterized by its effective curvature, or equivalently the inverse effective mass, at the touching point. For simplicity, we consider the effective masses along the principal axis $q_x = 0, q_y = 0$ ($m_p^\pm$) and the diagonal axis $q_x + q_y = 0, q_x - q_y = 0$ ($m_d^\pm$). The masses are

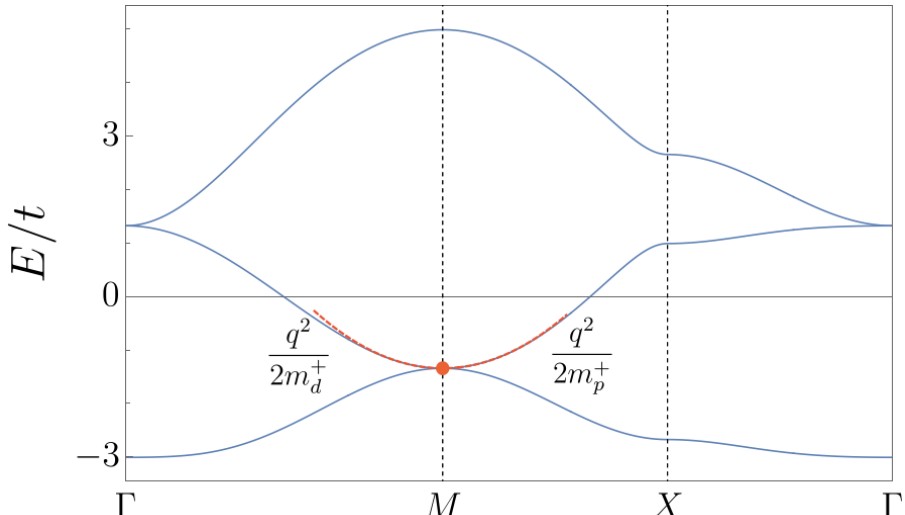

Figure 2: **Quadratic band touching**: The band structure of the three band model Eq. (3) in the strong hybridization limit. Two QBT points are present in the $\Gamma$ and $M$ points. We concentrate on the lower energy QBT at the $M$ point, indicated with a red dot, as there are no other bands at this energy (i.e. it is an isolated quadratic band touching). We define the curvature of the quadratic bands with the effective masses $m_{p/d}^{\pm}$, where the effective dispersion near QBT for the upper band is shown as dashed lines.

defined from the low energy dispersion

$$E^{\pm}(q_y = 0) = \pm \frac{|\mathbf{q}|^2}{2m_p^{\pm}}, \quad E^{\pm}(p_x = p_y) = \pm \frac{|\mathbf{q}|^2}{2m_d^{\pm}}. \tag{5}$$

The $\pm$ indicates the electron-like(+) and hole-like($-$) bands touching at the QBT point. Note that the $C_4$ symmetry of the system ensures the $m$'s are well defined with $p_x \leftrightarrow p_y$, $p_i \leftrightarrow -p_i$ in the definition. The $E_{\text{QBT}}$ and the quadratic dispersion described by $m_{p/d}^{\pm}$ are shown in Fig. 2 as a red dot and dashed lines.

To construct the full Hamiltonian of interest $H = H_0 + H_V$ we include a single particle potential:

$$H_V = \sum_{\mathbf{r}} \Psi_{\mathbf{r}}^{\dagger} V(\mathbf{r}) \Psi_{\mathbf{r}}, \tag{6}$$

where $\Psi_{\mathbf{r}}$ is the Fourier transform of $\Psi_{\mathbf{k}}$. We take $V(\mathbf{r})$ to be quasiperiodic with the underlying optical lattice

$$V(\mathbf{r}) = W[\cos(Qx + \phi_x) + \cos(Qy + \phi_y)], \tag{7}$$

with an incommensurate wave vector $Q$ (i.e., $Q/2\pi$ is an irrational number in the thermodynamic limit), $W$ is in units of $t$ throughout, and $\phi_{\mu} \in [0, 2\pi)$ is a random offset of the potential. We focus on the behavior of the model in the space of $W - Q$ and consider a few particular choices of incommensurate $Q$. These include taking the system size to be given by the $n$th Fibonacci number $L = F_n$ and the quasiperiodic wavevector to be $Q_L/2\pi = F_{n-2}/L$ such that as $L \to \infty$ we have $Q_L/2\pi \to [(\sqrt{5} + 1)/2]^{-2}$. We also focus on $Q_L/2\pi = F_{n-4}/L$ which corresponds to $Q_L/2\pi \to [(\sqrt{5} + 3)/2]^{-2}$ as $L \to \infty$. These $Q_L$'s are a finite system approximate for the true incommensurate $Q(\equiv \lim_{L \to \infty} Q_L)$ and we emphasize that the approximation is controlled, i.e. $|Q_L - Q|$ strictly decreases to 0 as $L$ increases, when $Q_L$'s are defined as above with the Fibonacci numbers.

To determine the properties of the model we use exact diagonalization and Lanczos to determine the eigenenergies $E_i$ and eigenstates $|E_i\rangle$. From these we determine the inverse participation ratio (IPR) in the basis $|\alpha\rangle$ (in particular we focus on $\alpha = \mathbf{r}$ and $\mathbf{k}$) that is given by

$$\mathcal{I}_\alpha(E) = \sum_\alpha |\psi_\alpha(E)|^4, \tag{8}$$

where $\psi_\alpha(E) = \langle\alpha|E\rangle$. If the wavefunction at energy $E$ is delocalized in the basis $\alpha$ then its IPR will go like $\mathcal{I}_\alpha(E) \sim 1/L^2$ whereas if it is localized it will approach an $L$ independent constant, i.e. $\mathcal{I}_\alpha(E) \sim$ const. On the other hand, if the wavefunction is critical then it will develop multifractal scaling that is characterized by $\mathcal{I}_\alpha(E) \sim L^{-\tau(2)}$ where $\tau(2)$ is the so-called second fractal dimension [46].

We study the effective band structure of the model in a mini Brillouin zone (mBZ) by twisting the boundary conditions by an amount $\theta = (\theta_x, \theta_y)$, which shifts the momentum $\mathbf{k} \to \mathbf{k} + \theta/L$. By treating the entire $L \times L$ system as a supercell, twisting allows us to access the Bloch momentum that live in a mBZ of size $2\pi/L \times 2\pi/L$. Thus, by determining the energy spectrum as a function of the twist $\{E_i(\theta)\}$ we obtain an effective dispersion in the mBZ. Our QBT of interest, at the $M$ point in the original Brillouin zone, is at the $\Gamma$ ($M$) point of the mBZ for an even (odd) $L$.

Note that there is no particle-hole symmetry in the bare model $H_0$ [Eq. (3)], and particularly at the QBT energy of interest. Therefore, the $E_{QBT}$ will not be stable as we include the potential term $H_V$ [Eq. (6)] and effectively tracking the QBT states and its energy as we tune the quasiperiodic potential is important. To achieve this, we compare the wavefunction overlap between the known QBT state at $W$, $|E_{QBT}(W)\rangle$, and states in the vicinity of $E_{QBT}$ at $W + \delta W$, $|E_i(W + \delta W)\rangle$. For the QBT state at $W + \delta W$, the overlap with $|E_{QBT}(W)\rangle$ will be significantly larger than the other states (Please refer to the Appendix B for details). The QBT energy depends on the random phases $\phi_\mu$'s and is computed separately for each sample of random phases.

We also compute the density of states (DOS),

$$\rho(E) = \frac{1}{L^2} \sum_i \delta(E - E_i), \tag{9}$$

using the kernel polynomial method (KPM) [47] by expanding $\rho(E)$ in a Chebyshev expansion up to an order $N_C$. We average over 100 samples with different $\phi_i$'s in the data shown in the main text. The low energy density of states in two dimensions in the vicinity of an isolated band crossing takes the form

$$\rho(E) \sim |E|^{2/z-1}, \tag{10}$$

where $z$ is the dynamical exponent that relates energy to length via $E \sim L^{-z}$. For a QBT, $z = 2$ results in a finite density of states at the QBT energy. In order to probe the low energy scaling of the DOS we will utilize the scaling with the KPM expansion order $N_C$ [15]. As a result of the finite expansion order of the KPM and the Jackson Kernel used here, the Dirac-delta functions in Eq. (9) are broadened to (approximately) Gaussians with a finite width $\delta E = \pi D/N_C$ where $D$ is the bandwidth. Thus, we can use the scaling with $N_C$ to determine the value of $z$ as Eq. (10) implies $\rho(E_{\text{QBT}}) \sim N_C^{1-2/z}$.

## 3 Renormalized Excitation Spectrum

To determine the phase diagram of the model as we tune the strength and moiré wavelength of the quasiperiodicity we start by computing the renormalized low energy excitation spectrum. The phases and transitions we identify are then corroborated as bona fide quantum

phase transitions through studying the nature of the wavefunctions in Sec. 4. We first study the nature of the low energy excitation spectrum in the vicinity of the QBT and how it is renormalized by the quasiperiodic potential. In order to assess these effects we use a combination of diagrammatic perturbation theory and numerical computations of the energy as a function of twisted boundary conditions.

## 3.1 Perturbation Theory

In this section, we use perturbation theory to analytically study the weak coupling regime ($W \ll 1$). Here, we use the full $\mathcal{H}_0(\mathbf{k})$ in Eq. (3) and include the quasiperiodic potential $\mathcal{H}_V$ as a perturbation using diagrammatic perturbation theory [14]. After formally performing the perturbative calculation, we expand our results near the QBT point up to second order in $\mathbf{q} \equiv \mathbf{k} - (\pi, \pi)$, and thus the resulting theory is only valid near the QBT point. This is sufficient to extract estimates of the stability of the QBT, the QBT energy, and the renormalized dispersion (i.e. effective mass) near the QBT point. For these purposes it is important that the QBT is isolated in the band structure and no other parasitic bands cross the Fermi energy at the QBT energy.

To focus on the energy of the QBT, we add a chemical potential $\mu = 2t_{pp} - 2t'_{pp}$ to the unperturbed Hamiltonian to shift the QBT to zero energy for convenience. This does not affect the perturbation theory itself, however, it allows us to expand also in the energy and get closed form solutions, e.g., Eq. (14) below. The chemical potential $\mu$ is a new parameter of the theory and renormalizes independently, although its bare value is related to other hopping parameters. Therefore, we use $\mathcal{H}_0(\mathbf{k}) + \mu \mathbb{1}_{3\times 3}$ as our final unperturbed Hamiltonian. We evaluate the single-particle self energy at second order, which yields the renormalized effective Hamiltonian up to second order in $\mathbf{q}$,

$$\tilde{\mathcal{H}}^{(2)}(\mathbf{q}) =$$
$$\begin{pmatrix} \tilde{t}_{dd}(4 - q_x^2 - q_y^2) + \tilde{\delta} + \tilde{\mu} & -2i\tilde{t}_{pd}q_x & -2i\tilde{t}_{pd}q_y \\ 2i\tilde{t}_{pd}q_x & \tilde{t}_{pp}(-2 + q_x^2) + \tilde{t}'_{pp}(2 - q_y^2) + \tilde{\mu} & \tilde{\alpha}q_x q_y \\ 2i\tilde{t}_{pd}q_y & \tilde{\alpha}q_x q_y & \tilde{t}_{pp}(-2 + q_y^2) + \tilde{t}'_{pp}(2 - q_x^2) + \tilde{\mu} \end{pmatrix},$$
(11)

where the tilde indicates the variables are renormalized relative to Eq. (4). Details of the calculation and the lengthy expressions for the renormalized parameters are given in Appendix C as their specific form are not of direct relevance to the discussion. From the perturbation theory and numerics we are able to identify magic-angles and construct the phase diagram shown in Fig. 3.

There are a few takeaways from Eq. (11). First, in the vicinity of the QBT, the perturbation theory preserves the structure of the Hamiltonian and only renormalizes the effective parameters. One exception is the $\tilde{\alpha}$ term, which is generated in the perturbative process, i.e., it can be viewed as being renormalized from a bare value of $\alpha = 0$. Therefore, the dispersion remains quadratic in general, except for the special points with so-called "magic angle condition" which we elaborate later.

Second, the touching of the two quadratic bands is stable. QBT appears as a double degeneracy at $\mathbf{q} = 0$, which is a feature remaining in Eq. (11). The QBT energy can be read from the diagonal Hamiltonian $\mathcal{H}^{(2)}(0)$ as

$$E_{QBT}^{(2)}(W, Q) = -2\tilde{t}_{pp} + 2\tilde{t}'_{pp} + \tilde{\mu}. \tag{12}$$

As mentioned earlier, the lack of particle-hole symmetry in the system implies that $E_{QBT}$ will change as the quasiperiodic potential is applied as shown in Eq. (12) and Fig. 4.

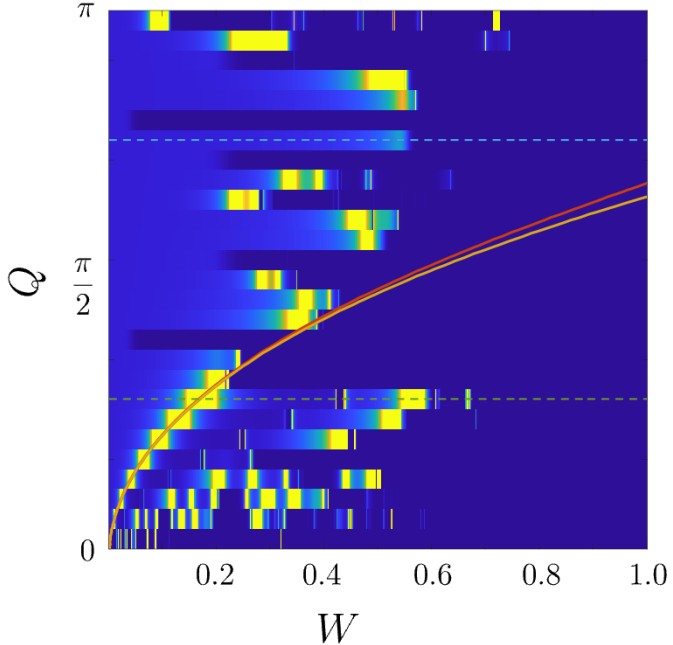

Figure 3: **Phase Diagram**: The perturbation theory predictions on diverging mass (solid lines) are plotted on top of the numerical calculation of the corresponding quantity that is a product of all effective masses at the QBT point ($\prod m_{p/d}^{\pm}$) computed from a $L = 55$ system size. The color scale yellow (blue) corresponds to large (small) $\prod m_{p/d}^{\pm}$ and the yellow region emerging from the origin matches very well with the perturbation results. The light blue regions correspond to QBT phases where as the yellow regions identify each magic-angle condition with a flat dispersion in part (or all) of the QBT. The precise location of each magic-angle will weakly shift with increasing $L$ but the yellow in the phase diagram is a good representative for the location of each magic-angle transition. The dark blue region (connected to $W \to \infty$) is the QBT broken phase where gap opens at the nodal touching point. The two values of $Q$'s we concentrate on the following discussion are also indicated as dashed lines.

We can calculate the change in effective masses (see Sec. 2 and Fig. 2 for definitions) after including the perturbation:

$$
m_p^+(W,Q) = \left[ 2\tilde{t}_{pp} - \frac{8\tilde{t}_{pd}^2}{4\tilde{t}_{dd} + 2\tilde{t}_{pp} - 2\tilde{t}_{pp}' + \tilde{\delta}} \right]^{-1},
$$

$$
m_p^-(W,Q) = \frac{1}{2\tilde{t}_{pp}'},
$$

$$
m_d^+(W,Q) = \frac{1}{\tilde{t}_{pp} - \tilde{t}_{pp}' - \tilde{\alpha}},
$$

$$
m_d^-(W,Q) = \left[ \frac{8\tilde{t}_{pd}^2}{4\tilde{t}_{dd} + 2\tilde{t}_{pp} - 2\tilde{t}_{pp}' + \tilde{\delta}} - (\tilde{t}_{pp} - \tilde{t}_{pp}' + \tilde{\alpha}) \right]^{-1}. \tag{13}
$$

Interestingly, the perturbative expressions show that it is possible for the renormalized masses to diverge, signalling the generation of flat bands. This is thus the natural extension of the concept of the "magic-angle condition" suitably generalized for Dirac semimetals [14] to the

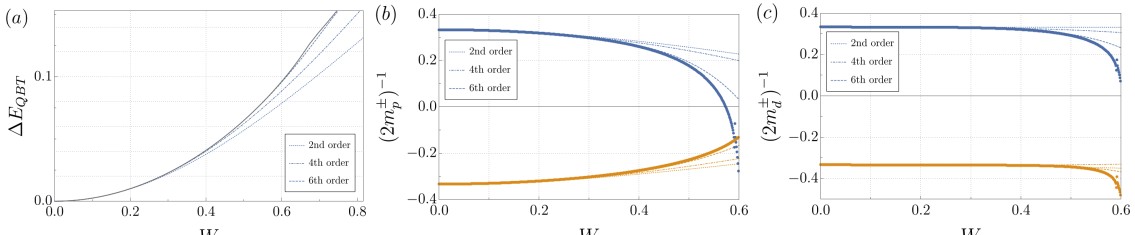

Figure 4: **Renormalization of the quadratic band touching**: Comparison of the perturbation theory (dashed lines) and numerical calculations for $L = F_{12} = 144$ and $Q_L/2\pi = F_{10}/L = 55/144$ system across the magic-angle transition. (a) The shift of the QBT energy as $W$ increases. Note that the QBT energy is not fixed due to lack of particle-hole symmetry. The QBT splits in the numerical calculation near $W \simeq 0.65$, and the energy difference is in the order of $10^{-3}$, which is not visible in this scale. (b,c) The effective curvature of the upper (blue) QBT band and lower (yellow) QBT band in the direction of (b) principal axis and (c) the diagonal direction. As shown in (b), the diverging effective mass $m_p^+$ around $W_c \approx 0.573$ identifies the magic-angle condition where part of the QBT has become flat.

QBT case. This is also consistent with the notion of a magic-angle in twisted double bilayer graphene in the absence of trigonal warping and particle hole asymmetric perturbations [40]. Thus, in the following whenever at least one of the effective masses diverges we refer to this as a magic-angle condition, which will also be accompanied by a significantly enhanced density of states at the QBT energy. In the limit of an incommensurate potential we show in Sec. 4 that each magic-angle coincides with a delocalization of eigenstates in momentum space and are thus in fact eigenstate phase transitions. We also find in the perturbation theory [Eq. (13)], and later confirm from the numerical calculations, that the four masses do not diverge simultaneously. In our parameter regime, $m_p^+$ diverges first which is immediately followed by the divergence of $m_p^-$. The masses along the diagonal, $m_d^\pm$, do not diverge before the second order perturbation theory breaks down.

The renormalized parameters have a complicated form that is not particularly illuminating and therefore finding a closed form for the magic angle condition ($m^{-1} = 0$) is formidable. However, for $m_p^-$ which has a relatively simple form, we can find the magic angle condition after expanding up to linear order in energy:

$$W_c(Q) = 2\left[\frac{2t_p' \sin^2 Q - t_{pQ}\cos Q}{2t_{pQ}^3} + \frac{2t_{pd}^4 \sin^2 Q - t_{pQ}(t_p' t_{dQ}^2 + t_d t_{pd}^2 \sin^2 Q)}{2t_{pQ}t_p'(t_{dQ}t_{pQ}' - t_{pd}^2 \sin^2 Q)^2}\right]^{-1/2}. \quad (14)$$

The $t_{pQ}$, $t_{pQ}'$, and $t_{dQ}$ are defined as follows:

$$t_{pQ} \equiv t_p - t_p'\cos Q - \mu/2,$$
$$t_{pQ}' \equiv t_p' - t_p\cos Q + \mu/2,$$
$$t_{dQ} \equiv t_d(1 + \cos Q) + (\delta + \mu)/2. \quad (15)$$

We plot the function of $W_c(Q)$ in Eq. (14) for which $[m_p^-(W,Q)]^{-1} = 0$ together with the numerically evaluated perturbative result for the previous magic angle condition $[m_p^+(W,Q)]^{-1} = 0$ in Fig. 3, as the two solid curves (that are indistinguishable at this scale at small $W$). Comparing the perturbative results with the exact numerical calculations of the model for a finite system size, which are described in more detail in the following section, we find that the second order perturbation theory predicts the magic angle condition rather accurately for small $Q$'s where the diverging effective mass occurs at a relatively small value of $W$.

For larger values of $Q$, second order perturbation is not enough and higher order corrections should be included to predict the correct phenomenology. To go beyond second order in the analytic perturbation theory is complicated, primarily due to the complex structure of the bare theory [Eq. (3)]. However, we can proceed to higher orders in perturbation theory numerically, by writing a tight-binding model in momentum space as in Ref. [22] (See Appendix D for details of this calculation). Using this numerical perturbation theory, we calculate the dispersions up to 6th-order, showing the results in Fig. 4.

Fig. 4(a) shows the energy of the QBT point $E_{QBT}$ (note that Eq. (12) is its expression in second order perturbation theory). The second, fourth, and sixth order perturbation theory results are compared with the numerically exact result for $L = F_{12} = 144$ and $Q_L/2\pi = F_{10}/L = 55/144$. One observes that the second order perturbation theory agrees well with the numerics for $W \ll 1$, and for large $W$'s the perturbation theory progressively approaches the numerical result as we get to higher-orders.

The masses $m_{p/d}^{\pm}$ are also calculated and compared with the same numerical simulations in Fig. 4(b,c). We see that the agreement with the numerics becomes significantly better as we include higher order corrections, and the 6th order result shows good agreement. Note that the relatively slow convergence to the numerical value in Fig. 4(b,c) are because we chose a large $Q(= 2\pi[(\sqrt{5} + 1)/2]^{-2})$ where important features occur for large values of $W$. For smaller $Q$'s lower order is sufficient, as it is evident from the comparison between numerics and second order perturbation theory in Fig. 3.

## 3.2 Numerical results

We now turn to numerically computing the low energy excitation spectrum that we compare to the perturbative results of the previous section. Going beyond the low energy renormalization near the QBT we also determine the nature of the formation of minibands and the nature of the density of states.

### 3.2.1 Renormalized dispersion

Now, we directly compute the single particle Hamiltonian $H = H_0 + H_V$ [Eq. (3),(6)] on a finite system. As mentioned in Sec. 2, we choose system sizes to be Fibonacci numbers to systematically approximate the irrational wavevectors. Calculating the energy eigenvalues with a twisted boundary condition (i.e., $\psi(\mathbf{r} + L\hat{u}) = e^{i\theta_\mu}\psi(\mathbf{r})$, where $\hat{u} = \hat{\mathbf{x}}, \hat{\mathbf{y}}$) is equivalent to considering the whole $L \times L$ system as a supercell, and thus we can calculate the energy dispersion in the folded-Brillouin zone labeled by twists $\theta_x$ and $\theta_y$. Starting from a system without a quasiperiodic potential, we increase the potential in small increments ($\Delta W = 0.001$) and obtain the eigenstates via Lanczos. Then, we track the QBT state by searching for the maximum overlap with the known QBT state in the previous $W$. This procedure is elaborated in Appendix B.

With the dispersion and knowing the QBT state, we can numerically obtain the quantities calculated by perturbation theory. The comparison between the QBT energy and effective masses from the two methods are already presented in Figs. 3, 4 which showed good agreement.

In Fig. 3, to determine if any of the masses have diverged the numerical data shown is the product of the four effective masses ($\prod_{a=p,d,b=\pm} m_a^b$) for a $L = F_{10} = 55$ system size. Since the four masses diverge at different points, we use this measure to indicate any band flattening in the two bands and two directions that help us identify each magic-angle transition. A line of magic-angle conditions (i.e. diverging effective mass) emerges from the origin, following the perturbation theory prediction. There is also a second line of band flattening occurring at a larger $W$ which is depicted as a dashed line. The mass divergence mentioned above occurs

very sharply while the mass changes its sign, and the system reenters the QBT phase after the divergence.

Another feature in Fig. 3 is the phase at large quasiperiodic potential, approximately $W \gtrsim 0.65$, displayed as dark blue. In this regime, the quasiperiodic potential is strong enough that gaps open up at the QBT. The system loses all its quadratic touching character and we call this a QBT broken phase. This gap opening can be explicitly seen from the calculation of $E_{QBT}$. The numerical data in Fig. 4(a) is actually split at large $W$, however the gap is small and the effect is not visible in this scale.

Considering the QBT broken phase, there are commensurate artifacts from the finite size in this figure. For finite size systems with periodic (and twisted) boundary conditions and $Q/2\pi = m/L$, special non-coprime ratios (of $m$ and $L$) can simply gap out the QBT. For example, the QBT broken state remains largely extended when $Q/2\pi = 1/2$ because for this $Q$ the potential is $V(\mathbf{r}) = W[\cos(\pi x + \phi_x) + \cos(\pi y + \phi_y)]$, which simply quadruples the unitcell (a factor of 2 from each directions) and nothing else. For these commensurate $Q$'s there is no delocaization in momentum space (as $\mathcal{I}_{\mathbf{k}}$ is then bounded from below due to Bloch's theorem [14]). Similar, but less prominent situations are observed in $Q/2\pi = 1/3, \ 1/4, \cdots$ as well. This is an artifact of the finite system we are simulating, and thus this feature will not be present in the thermodynamic, incommensurate $Q$ limit.

### 3.2.2 Minibands and the density of states

To assess the renormalized spectrum from across a broader energy range we compute the density of states $\rho(E)$. We expect the DOS will be enhanced when the effective mass diverges, and can also directly observe the gap formation from the DOS. Fig. 5(a) shows how $\rho(E)$ evolves as the quasiperiodic potential increases for $Q_L/2\pi = F_8/L$ and $L = F_{12} = 144$. From the upper panel, we observe a very small gap near $W = 0.14$ (the arrow near $E = -0.12$) that quickly vanishes, and for larger $W$ a clear gap is opened for $W = 0.22$. This creates a miniband with an enlarged unit cell (downfolded Brillioun zone) at low energy indicated by the dashed arrows. As we increase $W$ further in the lower panel, a second gap is opened inside the first miniband for $W = 0.49$ (arrow near $E = -0.06$) that becomes prominent for both positive and negative energies around $W = 0.52$ creating a second miniband with an even smaller mini Brillouin zone.

We can understand the origin of the gaps and minibands by investigating the number of states within the miniband. If the mBZ has an area of $A$, the number of states in the miniband near the QBT should be $\frac{2}{3}\frac{A}{4\pi^2}$. The 2/3 factor reflects that only two bands (which are quadratically touching at the QBT) out of the three orbitals contributes to the miniband and the later factor is the ratio of the mBZ to the full Brillouin zone. Considering the two minibands found in the $W = 0.52$ data, let us label the band roughly within $-0.05 \leq E \leq 0.05$ as the 1st miniband (denoted $MB_1$), and that within $-0.11 \leq E \leq 0.10$ as the 2nd miniband (denoted $MB_2$). By integrating the DOS in the first miniband $[\int_{MB_1} \rho(E)dE]$ we find that the $A_{MB_1} = Q^2$. Similarly, for the second miniband we find $A_{MB_2} = 2Q^2$. Thus, the quasiperiodic potential has "carved out" a mBZ whose size can be understood by examining scattering on the Fermi surface at a finite energy away from the QBT.

Let us consider a schematic Fermi surface as in Fig. 5(b). The circles represent the Fermi surfaces (larger circle has a larger Fermi energy) and the arrows are the quasiperiodic wavevectors $Q\hat{x}$ and $Q\hat{y}$. The red dots are all connected through a second order hopping process of either $Q\hat{x}$ or $Q\hat{y}$. All the parallel points in the inner dashed square are connected likewise. Through this second order process in the quasiperiodic potential scattering, gap forms at the inner dashed square, carving out a mBZ out of the full BZ. This mBZ precisely has the area of $Q^2$ and is the first mBZ seen in Fig. 5(a). The blue dots, and the parallel points in the outer

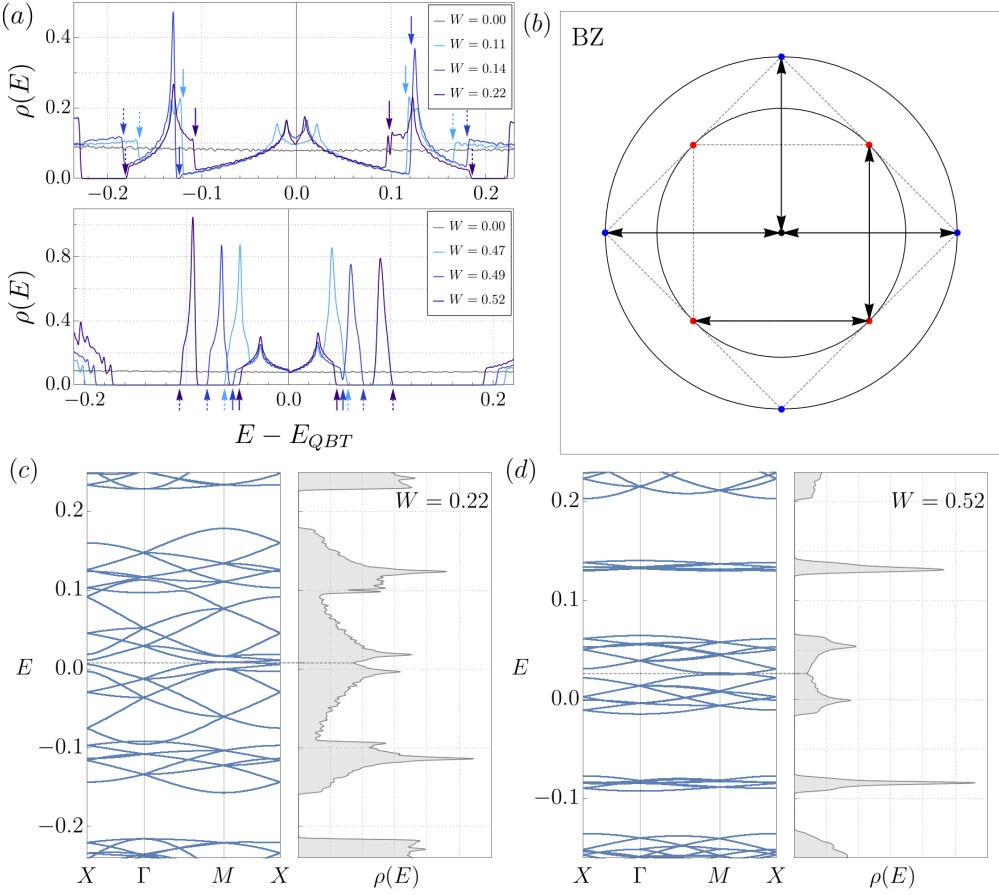

Figure 5: **Miniband formation and renormalized dispersion**: (a) DOS for different $W$ values with $Q_L/2\pi = F_{n-4}/L$, $L = F_{12} = 144$. The boundary of the first (solid) and second (dashed) minibands are shown in arrows. (b) The schematic figure of the Fermi surface and points connected via the quasiperiodic potential through second (red dots) and fourth (blue dots) order processes. The arrows are of length $Q$. (c,d) The DOS for $W = 0.22, 0.52$ are plotted side-by-side with the corresponding band structure, calculated for a $L = 21$ system.

dashed square, are similarly connected through a fourth order process of scattering in $Q$. The second mBZ in Fig. 5(a) is this outer square, which has the area $2Q^2$.

From this counting of states procedure, we can find where the miniband develops even before a clear gap opens up. In Fig. 5(a), we have indicated those points as solid (the first miniband from 2nd order process) and dashed arrows (the second miniband from the 4th order process). To sum up the information from the DOS and the counting of states, the gap opens at the negative energy first near $W = 0.14$ but quickly closes due to the density of states from the second miniband. A large gap separating the second miniband emerges shortly and persists, and the first miniband re-emerges as we increase the quasiperiodic potential to $W = 0.49$ and becomes very prominent around $W = 0.52$. In Fig. 5(c)(d), we plot the DOS computed with KPM and the band structure for a commensurate approximate wavevector $Q$ via twisted boundary conditions side-by-side to see how the first and second miniband emerges. The band structure was calculated for a system with $L = 21$ to clearly see the dispersions, which both show the QBT flattening (as expected based on the previous perturbation theory) in addition to the gap openings.

Another important piece of information we can get from the DOS is the dynamical exponent, see Eq. (10). The dynamical exponent for the QBT is $z = 2$ and this is expected to

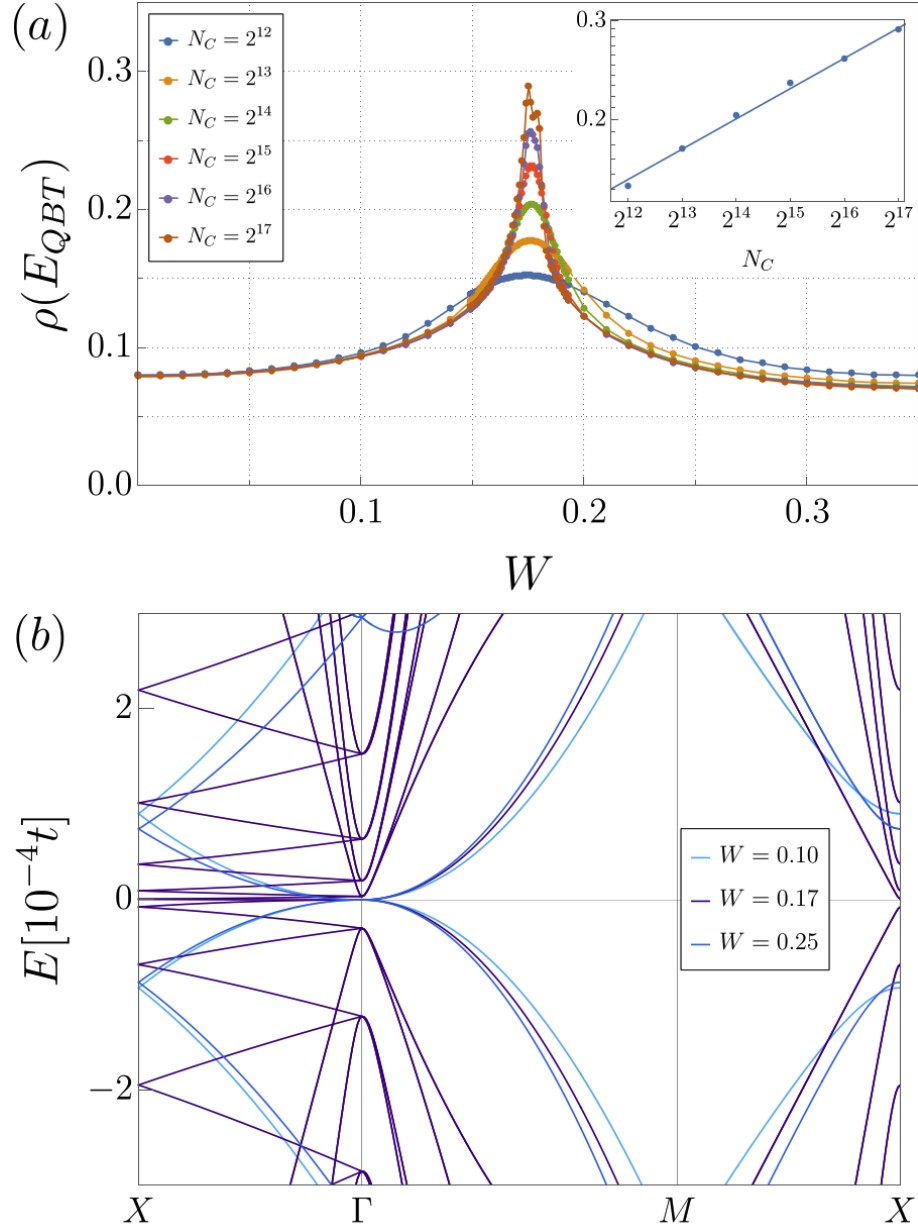

Figure 6: **Diverging density of states and flat bands at the magic-angle**: (a) The $\rho(E_{\text{QBT}})$ as a function of $W$ for a number of different $N_c$'s. The system size is $L = 144$ and the wavevector is $Q_L/2\pi = F_8/L$. The inset shows the peak value scales as a power law in $N_c$. (b) The dispersion of the same system for different $W$. This shows the system in QBT phase, enters a point ($W = 0.17$) where the band partially flattens, and re-enters the QBT phase.

increase near each magic-angle due to flat bands, resulting in an enhanced DOS per Eq. (10), e.g. any $z > 2$ will lead to a diverging low energy density of states. This enhancement can be best captured in its $N_C$ (KPM expansion order) scaling, where $\rho(E_{\text{QBT}}) \sim N_C^{1-2/z}$. In Fig. 6(a) we plot the $\rho(E_{\text{QBT}})$ as a function of $W$ for various $N_C$ values. We can clearly see that the $\rho(E_{\text{QBT}})$ is initially independent of $N_C$ and becomes enhanced and strongly $N_C$ dependent near the magic-angle transition near $W \sim 0.17$. As we increase the quasiperiodic potential further, the $N_C$ dependence disappears (at sufficiently large $N_C$) indicating that the system re-enters into a QBT phase.

In the inset we plot the maximum value of $\rho(E_{\mathrm{QBT}})$ for each $N_C$ in a log-log plot at the first magic-angle condition for this $Q$. The linear fit shows that $\rho(E_{\mathrm{QBT}}) \sim (N_C)^{0.179}$ for this critical point, giving $z \simeq 2.44$ and hence a diverging low energy density of states at the magic-angle. This is consistent with the prediction from the perturbation theory and the numerical effective mass calculation. If the quadratic term vanishes identically the next dominant term will be cubic and the exponent will increase to $z = 3$. However, we expect that for the first transition (or the first two very close transitions) only the $m_p^{\pm}$ diverges while the $m_d^{\pm}$ remains finite (see Fig. 4(b)(c), noting that the figures are at a different values of $Q$ but the qualitative behavior remains). Therefore the dynamical exponent should not increase to 3, but to some value between 2 (QBT) and 3 (cubic touching) which is precisely what we see from the $N_C$ scaling.

The position of the peak is not at the same $W$ value for the different $N_C$'s, however this is because there are actually two transitions happening as predicted from the perturbation theory. The two transitions corresponds to the diverging $m_p^{+}$ and $m_p^{-}$, respectively which also corresponds to the two lines in Fig. 3. Because the two transitions are very close in $W$, they cannot be resolved in Fig. 6(a) until $N_C$ is sufficiently large (e.g. $N_C = 2^{17}$).

We also calculated the band structure near the transition to explicitly verify this behavior. Fig. 6(b) shows the dispersion for a $L = 144$ system with quasiperiodic potential having values before, near, and after the transition. The QBT at the $\Gamma$ point shows clear quadratic dispersion for $W = 0.10$ (before transition). At $W = 0.17$, the system is close to the transition, and we can observe that the band is very flat along the $\Gamma - X$ line (where $m_p^{\pm}$ are defined) while it remains quadratic in the $\Gamma - M$ line (corresponding to $m_d^{\pm}$). And after the transition ($W = 0.25$) we see the quadratic dispersion restored in all directions and the band structure is very similar to that before the transition, hence we clearly identify a reentrant QBT phase.

## 4  Eigenstate transitions

So far we have studied the effect of the quasiperiodic potential on the spectrum of the model, and have shown how the band flattens and gaps open up to form minibands. Now we turn our focus to the nature of the eigenstates, and investigate any qualitative change on the wave-functions from the quasiperiodic potential. In particular, the phase diagram in Fig. 3 provides a clear picture on the phases and phase boundaries, however, a precise analysis requires connecting each phase to the properties of the underlying wavefunctions. It is now shown that the fundamental changes in the nature of the QBT point we have found are accompanied by transitions in the eigenstates in the incommensurate limit.

Guided by similar studies of Dirac semimetals [14, 15, 48] in incommensurate potentials we examine the IPR in real and momentum space. The stability of the QBT to quasiperiodicity implies a stable plane wave eigenstate at the QBT energy that is *localized* in momentum space, i.e., $\mathcal{I}_{\mathbf{k}}(E_{QBT}) \sim$ const. (see Eq. (8) for the defintion of the IPR). The transition out of this phase is then signalled by a delocalization of eigenstates in momentum space and $\mathcal{I}_{\mathbf{k}}(E_{QBT}) \sim 1/L^2$.

Fig. 7 is the plot of $\mathcal{I}_{\mathbf{k}}(E_{QBT})$, for a number of system sizes on two representative $Q$ values, $Q_L/2\pi = F_{n-4}/F_n$ and $Q_L/2\pi = F_{n-2}/F_n$, respectively. One common feature of the two $Q$'s are that $\mathcal{I}_{\mathbf{k}}(E_{QBT})$ is independent of $L$ for small $W$ signifying that the plane wave eigenstates at the QBT energy survive the quasiperiodic potential, hence this clearly demonstrates a stable QBT phase. However, $\mathcal{I}_{\mathbf{k}}(E_{QBT})$ becomes strongly $L$ dependent for large values of $W$. However, the $L$ dependence does not reach the scaling for a fully delocalized state and instead we find $\mathcal{I}_{\mathbf{k}}(E_{QBT}) \sim 1/L^x$ with $0 < x < 2$ signifying multifractal wavefunctions [14, 18, 46] and is thus not completely delocalized until $W \approx 0.9$. This can also be explicitly seen from the real space IPR (shown in the insets) which would have become independent of $L$ in the localized phase.

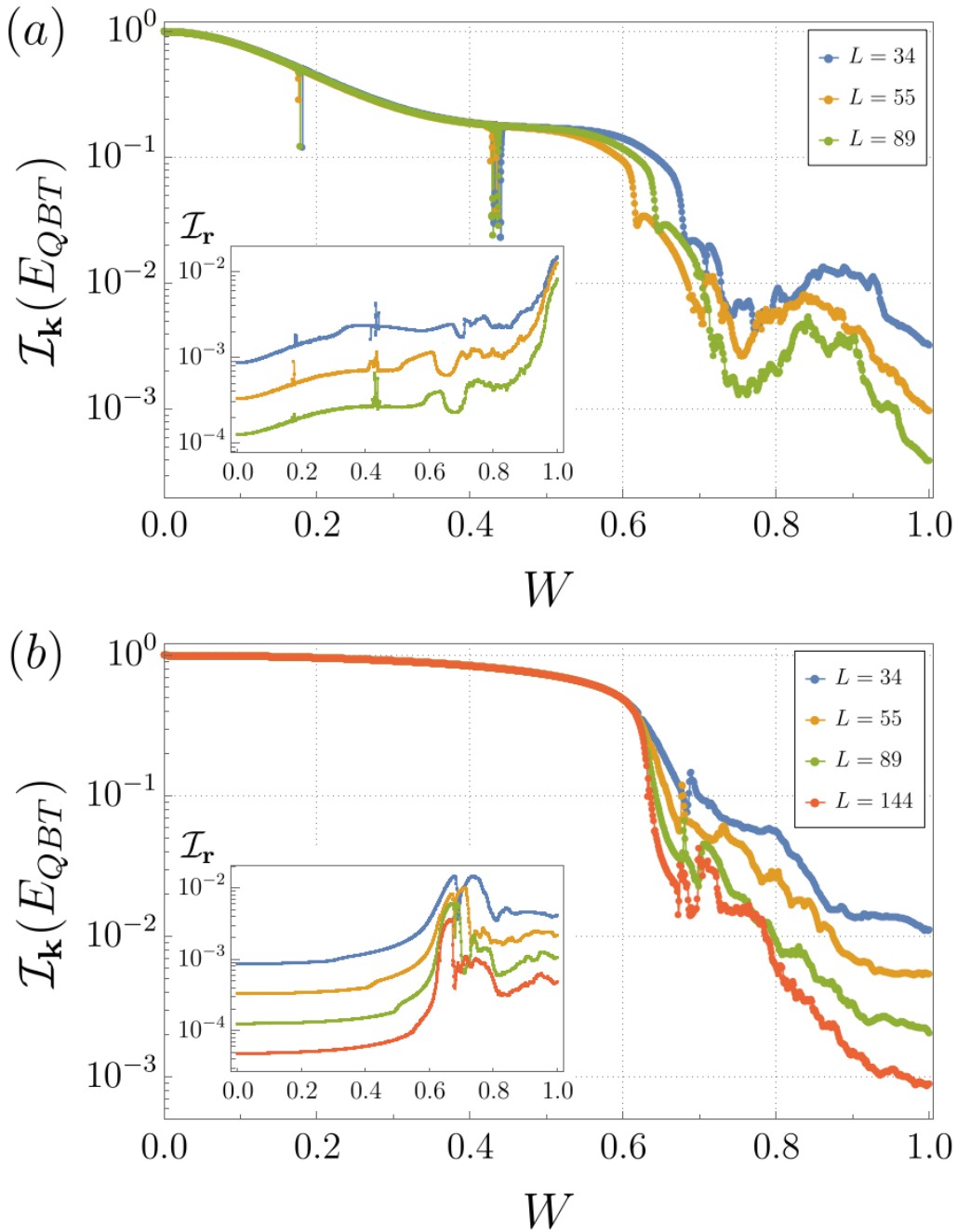

Figure 7: **Momentum space delocalization**: The scaling of IPR in momentum space (and real space shown in the insets) with the system size $L$ for wave vectors (a) $Q_L/2\pi = F_{n-4}/L$, (b) $Q_L/2\pi = F_{n-2}/L$. Sharp suppression of IPR is observed in (a), which coincides with the band flattening and magic-angle condition. Apart from these small regions, the IPR is independent of the system size prior to the main transition, indicating the QBT state is *localized* in the momentum basis. In contrast, the IPR decrease as the $L$ becomes larger, and the QBT state is *delocalized* in momentum space for large $W$. The insets are the real space IPR for the same parameters. The fact that the real space IPR does not reach the $L$-independent state infers that the system does not reach true localized phase in real space.

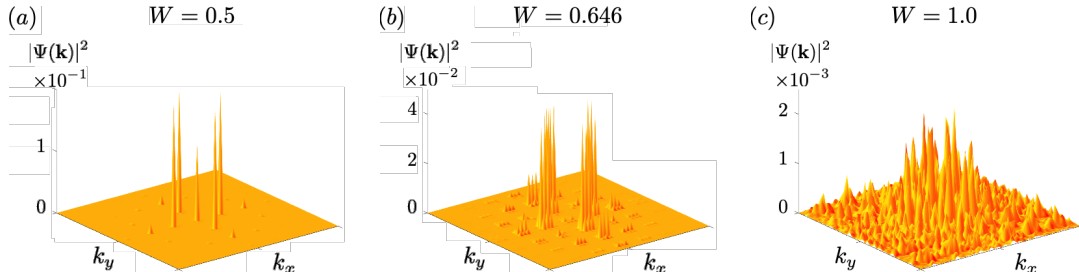

Figure 8: **Wave-functions**: The momentum space probability distribution, $|\Psi(\mathbf{k})|^2$, plotted as the system goes through the magic-angle delocalization transition in momentum space. The system size $L = 89 = F_{11}$ and the wavevector $Q_L/2\pi = F_7/F_{11}$ is used. The main peak at the $M$-point (center of the plot) and several satellite peaks are seen prior to the transition [(a)]; the peaks in (a) become strongly hybridized with states in the vicinity and starts to delocalize in momentum space [(b)]; the system further delocalizes and nears the fully delocalized state [(c)] in momentum space. Note the drastic difference in the $y$-scale.

Therefore, while strong $W$ will eventually localize the system [14, 15, 48–50], we do not focus on that regime here, which occurs for $W > 1$.

Let us take a closer look at the cut through the phase diagram with the rational approximate $Q_L/2\pi = F_{n-4}/L$ with $L = F_n$ [Fig. 7(a)]. First, we find two distinct transitions in the momentum space IPR, near $W \simeq 0.18$ and $W \simeq 0.43$, where $\mathcal{I}_\mathbf{k}(E_{QBT})$ develops strong $L$ dependence signifying the delocalization of the plane wave eigenstates in momentum space. Importantly, when we compare this to the changes in the spectrum (as shown in Fig. 3 and 4) we find that these transitions coincide with a diverging effective mass. Our results suggest that these are small metallic phases with delocalized wavefunctions, and the width of each phase grows with increasing $Q$. Thus, we have demonstrated that the magic-angle transitions in the QBT spectrum are in fact eigenstate phase transitions in the incommensurate limit. Upon passing through these delocalized phases, the model reenters a QBT phase, consistent with what we have found in the previous section: the system restores the quadratic dispersion shortly after band flattening, and thus the IPR in momentum space also becomes $L$-independent, reflecting the re-entrance to the QBT phase and stable plane wave eigenstates. We stress that all of these findings are consistent with phase diagram shown in Fig. 3.

Near $W \simeq 0.6$ with $Q_L/2\pi = F_{n-4}/L$, $\mathcal{I}_\mathbf{k}(E_{QBT})$ develops significant $L$ dependence and does not reenter the QBT phase. This is qualitatively different from the previous two IPR transitions that where accompanied by reentering the QBT phase. The earlier transitions are a result of only part of the QBT point becoming flat but some of the masses remain finite, in addition to Fig. 4 we show this clearly in the band structure in Fig. 6(b). However, for the transition at larger $W \approx 0.6$, the dispersion flattens in *all directions* and each of the four effective masses diverge. As a result, this moiré transition results in a complete destabilization of the QBT phase. We expect the same transition can in principle occur in twisted multilayer systems as well, however the value of $W$ in those systems (which is tunable via pressure [9]) is usually much smaller than $W_c$. However, we do expect this transition can be directly observed in the present context of an ultracold atom emulator, making these systems very rich due to the ability to turn off the interactions completely using a Feshbach resonance [51, 52].

At first glance, the $L$-dependence signifying the delocalization of the momentum space IPR near $W \approx 0.6$ with $Q_L/2\pi = F_{n-4}/L$ in Fig. 7 (a) is counterintuitive as it is non-monotonic in system size. However, this trend can be straightforwardly understood by considering the sequence of system sizes (equal to Fibonacci numbers) that we have considered for this $Q_L$. In particular, we see that the $L = F_{10} = 55$ system decrease first, and $L = F_{11} = 89$ and

$L = F_9 = 34$ follows. For each $L$, the transitions occurring at different potential strength for different system size naturally follows from approximating $Q$ as $Q_L$. Although $Q_L$ is an approximation successively approaching $Q$, the sign of $Q_L - Q$ alternates. For the $L = F_9, F_{10}, F_{11}$ considered in Fig. 7(a), we see the sequence of $Q_{F_{10}} < Q < Q_{F_{11}} < Q_{F_9}$. This is exactly the sequence we observe the suppression of IPR, and we can expect that in the thermodynamic and incommensurate limit, the transition will occur between the $L = 55$ and $L = 89$ transitions. Note that while the transition is very sharp in the small $W$ regime, the previous two transitions also follows the same sequence. In the perturbative sense, the first transition is of the lowest order and thus the deviation is not large but it becomes slightly more spread out in the second transition. The final transition is of the highest order among the three and shows the most prominent deviation. [1]

Turning to the cut with incommensurate wavevector $Q_L/2\pi = F_{n-2}/F_n$ [Fig. 7(b)], the system shows similar behavior to that of $Q_L/2\pi = F_{n-4}/F_n$, but for this parameter there is only a single magic-angle transition where all the effective masses diverge and there is no re-entrant phase at this larger value of $Q$. This data also clearly shows the multifractal scaling of the momentum sapce IPR when the wavefunctions delocalize in momentum space as we have $\mathcal{I}_\mathbf{k}(E_{QBT}) \sim 1/L^x$ with $0 < x < 2$ until $W \gtrsim 0.9$

To look directly at the qualitative aspects of the delocalization transition in momentum we show the momentum space probability density of the wavefunction, $|\Psi(\mathbf{k})|^2$, in Fig. 8. For small $W$, $|\Psi(\mathbf{k})|^2$ has a single prominent peak at the $M$-point (center of the figure) that signifies the stable QBT point and the satellite peaks are a perturbative effect that are connected to the $M$ point by "hops in momentum space" due to the $Q\hat{\mathbf{x}}$, $Q\hat{\mathbf{y}}$ vectors. In Fig. 8(a), we see that $W$ is large enough that the first satellite peaks became larger than the center peak at the $M$ point, and the second and third satellite peaks are also visible. However, when the system approaches the momentum space delocalization transition in Fig. 8(b), the peaks start to strongly hybridize with nearby momentum states and eventually delocalize in momentum space in a non-trivial manner, consistent with the scaling we observe in $\mathcal{I}_\mathbf{k}(E_{QBT}) \sim 1/L^x$ with $0 < x < 2$. After further increasing the potential, the wavefunction is completely delocalized in momentum space, as seen in Fig. 8(c), which is also where we find $\mathcal{I}_\mathbf{k}(E_{QBT}) \sim 1/L^2$. Note the difference in the $y$-scale in the three figures. From the momentum space wavefunctions, we were able to qualitatively observe the momentum space delocalization transition which was suggested from the IPR analysis.

## 5 Discussion

The various phases we have found in this manuscript, we expect, can be observed using existing experimental techniques for ultra cold atoms. The presence of magic-angles with re-entrant phases can be observed through wave-packets slowing down and speeding back up [14]. In addition, the miniband formation and flat bands can be observed in any spectroscopic signature that can be probed using band mapping techniques [53] or two photon Raman spectroscopy [54] to measure the dispersion as well as momentum-resolved radiofrequency spectroscopy to measure the spectral function [55]. In addition, the fundamental change in the eigenstates is expected to naturally appear in time of flight imaging and Bragg spectroscopy [19]. The presence of a harmonic trap is expected to introduce an additional length scale into the problem that will round out the magic-angle transitions into cross overs. These effects can be circumvented though via the use of box traps [56]. One important in-

---

[1] We note that $F_{n-4} = 21$ is not a co-prime with $L = 144$, and thus $Q_L/2\pi = F_{n-4}/F_n$ for this system is merely a nine copies of a $L = 48$ system. Therefore its result reflects a smaller system than the $L = 89$ one, and thus not included Fig. 7(a).

gredient of this realization is the high chemical potential (need to realize a stable filling of 3 atoms per site to fill the bands up to the QBT) for the optical lattice realization described near Eq. (A.1). While we expect this to be experimentally challenging, e.g. due to loss, we do not expect this to be detrimental to the proposal and hope it motivates further developments in this area.

As noted previously, the model under investigation has no particle hole symmetry (even on average) and thus the location of the QBT energy moves with increasing $W$, as shown explicitly in Fig. 4 (a). This has an effect that the $E_{QBT}$ changes with the quasiperiodic potential and complicates the numerical calculation. However, in cold atom experiments, this is not that much of an obstacle. The important quantity is the fraction of filling below the QBT point. Up to the first transition we observe that there are no bands crossing the $E_{QBT}$ energy, and thus the filling fraction is fixed to 1/3. After the first transition, the fraction changes, as there are many bands moving up and down across the $E_{QBT}$ value. Even in this case, the filling fraction can be easily computed from the DOS calculation and the experiments can also probe the appropriate QBT physics by starting from the respective filling.

It will be interesting to explore the role of short interactions (which can be tuned via a Feshbach resonance [51,52]) on the formation of symmetry broken states in the present setting. The formation of flat bands at each magic-angle condition with the large enhancement of the density of states are expected to greatly increase the value of the effective onsite interaction. As a result, we expect that in the vicinity of each magic-angle condition, interaction effects will dominate and drive the formation of correlated many body states. If such a symmetry broken phase gaps out the quadratic band touching we expect that this will realize topological quantum anomalous Hall phases [57,58].

## 6 Conclusion

In this work we have generalized the notion of magic-angles in Dirac semimetals to the case of quadratic band touching to emulate the physics of twisted double bilayer graphene. Our work has uncovered a series of magic-angle transitions where either part or the entire nodal point becomes flat with a dramatically renormalized band structure that lives on an effective moiré superlattice. These magic-angle transitions coincide with a wavefunction delocalization transition in the incommensurate limit, demonstrating this physics is universal. It will be very interesting to explore this connection to even higher order touching points, such as cubit or quartic, where our work suggests that magic-angle effect should survive in each case (though it may manifest itself in a slightly different fashion).

## Acknowledgements

We thank Jennifer Cano, Shiang Fang, Eslam Khalef, Elio König, Daniele Guerci, and Justin Wilson for useful discussions. This work is partially supported by the Air Force Office of Scientific Research under Grant No. FA9550-20-1-0136 and the Alfred P. Sloan Foundation through a Sloan Research Fellowship. The authors acknowledge the research computing resources that have contributed to the results reported here: the Office of Advanced Research Computing (OARC) at Rutgers, The State University of New Jersey (http://oarc.rutgers.edu), for providing access to the Amarel cluster.

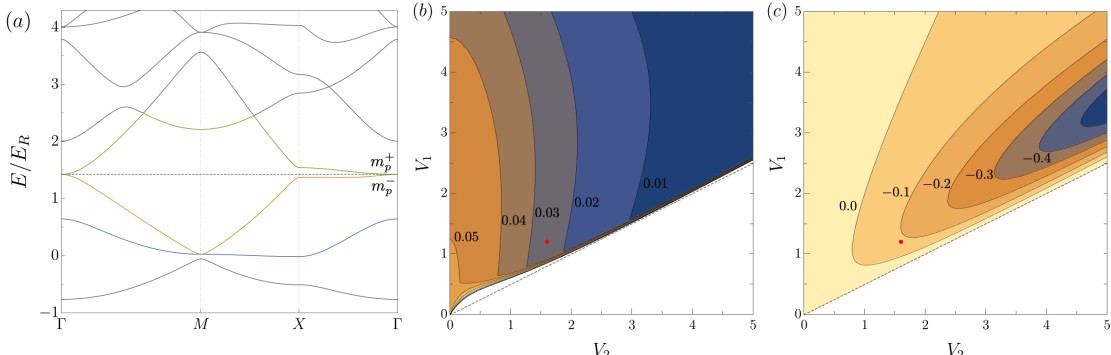

Figure 9: (a) The band structure obtained from the optical lattice potential [Eq. (A.1)], and plane wave basis of the Bloch wavefunction [36] for $V_1 = 1.6E_R$, $V_2 = 1.2E_R$. The three band tight-binding model in the main text corresponds to the three colored band. The chemical potential required for to reach the QBT at 3 atoms per site is depicted as a dashed line. The contour plot shows the value of the inverse effective mass (b) $(2m_p^+)^{-1}$ and (c) $(2m_p^-)^{-1}$ along the $\Gamma - X$ line as a function of $V_1, V_2$ (corresponding bands are indicated in (a)). We indicate the proposed parameters $V_1 = 1.6E_R$, $V_2 = 1.2E_R$ as a red dot, and the dashed line is the $V_2 = V_1/2$ line.

## A Experimental realization

The tight-binding model we describe in Sec. 2 can be realized in experiment with a Fermi gas composed of either $^6$Li or $^{40}$K. As proposed in Ref. [36], an optical lattice that can give rise to the multiorbital structure emerges from the potential,

$$V(x, y) = -V_1[\cos(kx) + \cos(ky)] \\ + V_2[\cos(kx + ky) + \cos(kx - ky)]. \tag{A.1}$$

The values of $V_1$ and $V_2$ determine the structure of the optical lattice and the strength of the hopping parameters $t_\alpha$ which are given by the overlap of the orbitals. Here, we need $V_2 > V_1/2$ to have potential minima at the bond centers [36].

In addition to requiring $V_2 > V_1/2$ the parameters we suggest to realize a QBT are $V_1 = 1.6E_R$, $V_2 = 1.2E_R$ with a filling of 3 per site for obtaining the QBT point at $\Gamma$, and $E_R = \hbar^2 k^2/4m$ is the recoil energy. (Note that the Alkali elements we have proposed are spinful) Fig. 9(a) shows the band structure of the optical lattice with the proposed parameters $V_1 = 1.6E_R$, $V_2 = 1.2E_R$. The colored second to fourth bands consist the three orbitals in the tight-binding model (Eq. (3)).

The reason we suggest filling of 3 per site rather than 2 (which corresponds to the low-energy QBT in the tight-binding model) is clear from this band structure, that is the touching point at $M$ is not a QBT. The second band in the $M - \Gamma$ line is dispersing upwards in energy, and this feature remains within a reasonable range of $V$'s ($0 \le V_1, V_2 \le 10E_R$). This is in contrast with the high-energy QBT at the $\Gamma$ point, where the dashed line indicates chemical potential to realize the QBT. Within the tight-binding model we chose the low-energy QBT without loss of generality, but this is not the case in optical-lattice realization.

Another feature to have in mind in choosing the parameters is the initial effective mass of the QBT. As one observes in Fig. 9(a), the $m_p^\pm$ along the $\Gamma - X$ line is much larger than $m_d^\pm$. Since the experimental signature of band flattening will be most clear when we start from a QBT with reasonably small effective mass, we find the values of $V_1, V_2$ which gives small $m_p^\pm$. Fig. 9(b,c) shows $(2m_p^\pm)^{-1}$ as a function of $V_1$ and $V_2$, and can see $V_1 = 1.6E_R$, $V_2 = 1.2E_R$

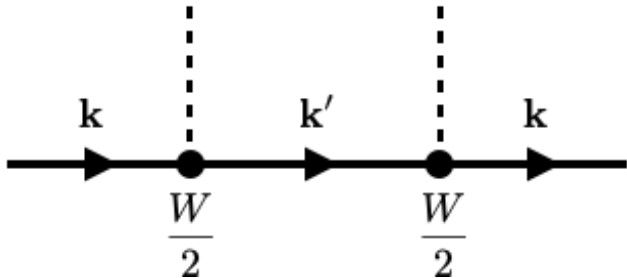

Figure 10: The Feynman diagram to calculate the second order self-energy [Eq. (C.3)]. The solid lines are the bare Fermion propergators, and the dashed lines are the quasiperiodic potential.

is a reasonable suggestion for the parameters. Note that the $m_d^{\pm}$'s are in general smaller in the parameter regime where $m_p^{\pm}$ is small, and therefore are less important in choosing the parameters.

For the quasiperiodic potential, at most, two pairs of additional lasers for each direction are needed. The $W$ will be determined by the laser intensity, and $Q$ by its wavelength. Qualitatively, the $W$ and $Q$ parameters are analogous to the interlayer tunneling strength and the twist angle of the twisted bilayer systems, respectively [14].

## B    Calculation of QBT states for $W > 0$

The two QBT points from the bare Hamiltonian [Eq. (3)] are at $\Gamma$ and $M$ points. Eq. (3) is diagonal at those momentum and can easily verify the QBT energies are $\pm 2(t_{pp} - t'_{pp})$.

Now let us consider a finite system of $L \times L$ and concentrate on the $M$ point with QBT energy $-2(t_{pp} - t'_{pp})$, which is lower of the two within our parameters of interest. Since we exactly know the QBT energy for $W = 0$, we can use Lanczos to calculate the QBT state $|E_{QBT}(W = 0)\rangle$. Let us assume we know $|E_{QBT}(W_0)\rangle$ for some $W_0$. We can again use Lanczos to calculate $n$ eigenstates $|E_i(W_0 + \delta W)\rangle$ whose energy is closest to $E_{QBT}(W_0)$. The $|E_{QBT}(W_0 + \delta W)\rangle$ will be the state with maximum overlap $\langle E_{QBT}(W_0)|E_i(W_0 + \delta W)\rangle$, for sufficiently large $n$ and small $\delta W$. We can obtain the QBT state for an arbitrary $W$ by induction, starting from $|E_{QBT}(W = 0)\rangle$.

If $|E_{QBT}(W_0)\rangle$ and $|E_{QBT}(W_0 + \delta W)\rangle$ are adiabatically connected, perturbation theory would suggest $\langle E_{QBT}(W_0)|E_i(W_0 + \delta W)\rangle \sim 1 - \mathcal{O}(\delta W)$ for small $\delta W$. However, note that since the QBT point is doubly degenerate the numerically obtained two states may not be adiabatically connected. In the extreem case of equal superposition $\langle E_{QBT}(W_0)|E_i(W_0 + \delta W)\rangle \sim 1/\sqrt{2} - \mathcal{O}(\delta W)$. During the process of finding the $|E_{QBT}(W)\rangle$ we check whether the overlap is greater than a certain value (for instance, 0.6) to assure the validity of the calculation.

## C    Analytical perturbation theory

In this appendix, we provide the details of the perturbation theory performed to calculate the effect of the quasiperiodic potential [Eq. (6)] in the vicinity of the QBT. We define the bare (non-interacting) Greens function of the fermions as:

$$G_0(\omega, \mathbf{k}) = \frac{1}{\omega - \mathcal{H}_0(\mathbf{k})}, \tag{C.1}$$

where $\mathcal{H}_0$ is Eq. (3). The dressed (interacting) Greens function is written as:

$$G(\omega, \mathbf{k}) = \frac{1}{\omega - \mathcal{H}(\mathbf{k})}, \tag{C.2}$$

where $\mathcal{H}$ is now the full Hamiltonian, including the potential term (Eq. (6)). We use the Dyson's equation $G(\omega, \mathbf{k})^{-1} = \omega - \mathcal{H}_0(\mathbf{k}) - \Sigma(\omega, \mathbf{k})$ where $\Sigma$ is the self-energy, and expand around the $M$-point where the QBT of interest is located (see Fig. 2). Up to second order perturbation theory, the self-energy can be expanded as:

$$\Sigma^{(2)}(\omega, \mathbf{k}) = \left(\frac{W}{2}\right)^2 \sum_{\pm, \hat{\mu} = \hat{\mathbf{x}}, \hat{\mathbf{y}}} \frac{1}{\omega - \mathcal{H}_0(\mathbf{k} \pm Q\hat{\mu})}. \tag{C.3}$$

The Feynman diagram corresponding to this process is shown in Fig. 10.

Calculating the diagram and also expanding the momentum up to second order in $\mathbf{q}$ where $\mathbf{k} = (\pi, \pi) + \mathbf{q}$, the self-energy is:

$$\Sigma^{(2)}(\omega, \mathbf{k}) = \omega \begin{pmatrix} \eta_8 & 0 & 0 \\ 0 & \eta_3 & 0 \\ 0 & 0 & \eta_3 \end{pmatrix} + $$

$$\begin{pmatrix} -\eta_9(4 - q_x^2 - q_y^2) + \eta_7 + 4\eta_9 & -2i\eta_1 q_x & -2i\eta_1 q_y \\ 2i\eta_1 q_x & \eta_4(q_x^2 - 2) - \eta_5(2 - q_y^2) + \eta & \eta_6 q_x q_y \\ 2i\eta_1 q_y & \eta_6 q_x q_y & \eta_4(q_y^2 - 2) - \eta_5(2 - q_x^2) + \eta \end{pmatrix}, \tag{C.4}$$

where $\eta = (2\eta_4 + 2\eta_5 + \eta_2)$. The $\eta$'s can be derived from Eq. (C.3), however the exact expressions are very complicated. To show a relatively simple expression, $\eta_8$ can be written as:

$$\eta_8 = -\frac{W^2 \left( (2t'_{pp} + \mu - 2t_{pp}\cos Q)^2 + 4t_{pd}^2 \sin^2 Q \right)}{\left( (2t_{dd}(1 + \cos Q) + \delta + \mu)(2t'_{pp} + \mu - 2t_{pp}\cos Q) - 4t_{pd}^2 \sin^2 Q \right)^2}. \tag{C.5}$$

The renormalized parameters in Eq. (4) are expressed in terms of $\eta$'s.

$$\tilde{t}_{dd} = \frac{t_{dd} - \eta_9}{1 - \eta_8},$$

$$\tilde{t}_{pp} = \frac{t_{pp} + \eta_4}{1 - \eta_3},$$

$$\tilde{t}'_{pp} = \frac{t'_{pp} - \eta_5}{1 - \eta_3},$$

$$\tilde{t}_{pd} = \frac{t_{pd} + \eta_1}{\sqrt{(1 - \eta_8)(1 - \eta_3)}},$$

$$\tilde{\delta} = \frac{\delta + \eta_7}{1 - \eta_8} + (\mu + \eta_2)\left(\frac{1}{1 - \eta_8} - \frac{1}{1 - \eta_3}\right),$$

$$\tilde{\mu} = \frac{\mu + \eta_2}{1 - \eta_3},$$

$$\tilde{\alpha} = \frac{\eta_6}{1 - \eta_3}. \tag{C.6}$$

# D   Numerical perturbation theory

In our theory, the perturbation $H_V$ [Eq. (6)] consists of two terms with definite momentum $Q\hat{\mathbf{x}}$ and $Q\hat{\mathbf{y}}$. Therefore, for a specific $\mathbf{q} = (q_x, q_y)$, one can numerically calculate higher order results of perturbation theory by solving the momentum space tight-binding model.

To illustrate this method, let us take the example of the second-order perturbation. If at most second order processes are allowed, the momentum that can be connected with $\mathbf{q}$ through $H_V$ are $\mathbf{q} \pm Q\hat{\mathbf{x}}$ and $\mathbf{q} \pm Q\hat{\mathbf{y}}$, and the matrix elements between those momentum are identically $(W/2)\mathbb{1}_{3 \times 3}$. Therefore the second-order perturbation theory can be described by the following $5 \times 5$ block Hamiltonian:

$$
\mathcal{H}_{\mathbf{q}} = \begin{pmatrix}
\mathcal{H}^{(2)}(\mathbf{q}) & \mathcal{W} & \mathcal{W} & \mathcal{W} & \mathcal{W} \\
\mathcal{W} & \mathcal{H}^{(2)}(\mathbf{q}+Q\hat{\mathbf{x}}) & 0 & 0 & 0 \\
\mathcal{W} & 0 & \mathcal{H}^{(2)}(\mathbf{q}-Q\hat{\mathbf{x}}) & 0 & 0 \\
\mathcal{W} & 0 & 0 & \mathcal{H}^{(2)}(\mathbf{q}+Q\hat{\mathbf{y}}) & 0 \\
\mathcal{W} & 0 & 0 & 0 & \mathcal{H}^{(2)}(\mathbf{q}-Q\hat{\mathbf{y}})
\end{pmatrix}, \qquad \text{(D.1)}
$$

where $\mathcal{W} = (W/2)\mathbb{1}_{3 \times 3}$. The eigenstates of this Hamiltonian, which are smoothly connected to the unperturbed eigenstates in $W \to 0$ limit are the exact states from second-order perturbation theory.

Generalizing this method to $2n$-th order perturbation theory is straightforward. There will be $2n^2 + 2n$ terms that are connected via $2n$-th order process of $H_V$ and the Hamiltonian $\mathcal{H}_{\mathbf{q}}$ will be a $(2n^2 + 2n + 1) \times (2n^2 + 2n + 1)$ block matrix. Identifying the momentum (placing the unperturbed Hamiltonian at the diagonal) and placing the $\mathcal{W}$ at proper off-diagonal positions give $\mathcal{H}_{\mathbf{q}}$ and diagonalizing it results in the $2n$-th order perturbation theory.

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
