# Peer review of "Emulating twisted double bilayer graphene with a multiorbital optical lattice"

_SciPost Physics, doi:SciPost Phys. 13, 033 (2022)_

## Round 1 · Referee Report · Anonymous (Referee 1) · 2022-4-20

Report

The authors propose to use a square optical lattice with 3 orbital per site, namely $p_x, p_y, d_{x^2-y^2}$, to emulate the quadratic band touching of Bernal stacked bilayer graphene. With the inclusion of quasiperiodic potential, they successfully identify the miniband structure and the magic-angle condition. Overall, the following questions should be addressed before we can recommend the publication of the work.

  1. As a guidance for the cold atom experiment, if there some experimental setup (which chemical element, how the hopping is tuned, how the potential is tuned) in the authors' mind that can potentially realize some of the points in the magic-angle condition. If so, what is the experimental signature to detect them.

  2. We know that, the strong electronic correlation within flat band gives birth to most of the intriguing physics, e.g. superconductivity, quantum anomalous Hall, in moire systems. To tackle such correlated physics, how can the correlation between cold atoms be encoded? And will the inclusion of the correlation change the magic-angle condition?

  3. The magic-angle condition is characterized by the Q and W. How is the condition related to the twisted angle in realistic system?

  4. In Fig.4, the authors show higher-order perturbation effective masses are closer to the numerical exact results. Can the author also show that, how the perturbation theory predictions (solid lines) get improved with the higher-order results?

  • validity: -
  • significance: -
  • originality: -
  • clarity: -
  • formatting: -
  • grammar: -

Author:  Junhyun Lee  on 2022-05-24  [id 2509]

(in reply to Report 1 on 2022-04-20)
Category:
remark
answer to question

  1. For the chemical element, we propose Fermi gases composed of either $^6$Li or $^{40}$K as canonical candidates. They are widely used in Fermion cold atom experiments with well-known cooling procedures and nice Feshbach resonance, in case one considers including interactions. The hopping parameter $t$'s are determined by the orbital overlaps between neighboring sites. Therefore, they are tuned by the depth of the underlying optical lattice. In the response to Referee report 2, comment 1, we suggest a range of values for the amplitudes of the lasers creating the optical lattice. There should be two pairs of additional lasers (for each direction) for the quasiperiodic potential. The strength of the potential will be determined by the laser intensity, and its wavevector by the laser frequency. For the quasiperiodicity, the laser frequency should be incommensurate with the optical lattice laser frequencies. The experimental signatures to detect the magic-angle condition are elaborated in the first paragraph of Section V. These include observation of wave-packet slowing down, band mapping techniques, two-photon Raman spectroscopy, and time of flight imaging. For more details on each method, please refer to the references in Section V. We included the additional information on experimental realization in Appendix A.

  2. While the moire systems have many interesting features of their own, it is indeed true that the strong correlation phenomena emerging from the flat bands (and thus the enhanced interaction) are one of the main reasons that moire systems are gaining such interest nowadays. Our work concentrates prerequisite of those phenomena, namely the underlying band structures and wavefunction properties of moire systems, and does not include interactions and the resulting instabilities. However, interactions can be tuned in the proposed cold-atomic system via a Feshbach resonance, and the $^6$Li and $^{40}$K atoms we have suggested above are species whose interactions are relatively easy to tune. In the last paragraph of Section V, where we discuss some implications of interactions, we included that the Feshbach resonance can be used to tune the interaction strength. The interplay between correlation and magic-angle condition is an interesting, and not fully resolved question. At the mean field, or Hartree Fock level, the current understanding is no, it does not shift the magic-angle, e.g. arXiv:2112.14752. Instead, relaxation of the twisted systems tends to have the most dominant effect in shifting the magic-angle.

  3. The $Q$ and $W$ in our theory corresponds to the angle and interlayer tunneling strength, respectively. To elaborate, for each lattice, twisting two layers with a certain angle will generate the moire pattern. The wave vector of the pattern will be the corresponding $Q$. Also, the interlayer tunneling strength, determined by the interlayer distance is directly related to the strength of the quasiperiodic potential $W$. This analogy is also presented in Ref. 14. In condensed matter systems, the interlayer tunneling is hard to control (unless with applied pressure) and thus most of the experiments control the condition with the relative angle. However, the theory does not have this restriction and we get a line of magic-angle conditions in the $Q-W$ plane. Cold-atom experiments can control $W$ with the intensity of the quasiperiodic potential laser, and $Q$ by using laser with different frequencies. To clarify this point, we add the above information at the end of Appendix A.

  4. We are unsure of what the referee means by the ‘perturbation theory predictions (solid lines).’ The solid line in Fig. 4(a) and points (which look like thick solid lines) in Fig. 4(b,c) are both numerical calculations, and independent of the order of perturbation theory. Instead, the numerical calculations get improved with system size, as we approach the incommensurate limit by increasing the system size. If the referee is asking about how perturbation theory gets improved by including higher-order results, Fig. 4 is demonstrating exactly that point, as we show perturbation theory with different orders and how it compares with the numerical results which we consider very close to the exact result. If the referee is asking about how numerical calculations get improved with system size, the system size dependence is shown for IPR (Fig. 7) but is not for QBT energy or masses. However, we have calculated band structures for many different system sizes ($L=144$ being the largest) and observe that increasing the size, say from $L=34$ to $L=144$, has less effect than increasing the perturbation order from 2 to 6. (Note that for a systematic approximation of the incommensurate potential, we restrict ourselves to $L$ being taken from the Fibonacci numbers). Therefore, we conclude that for the properties of the QBT point, $L=144$ is already large enough at this energy resolution.

---

## Round 1 · Referee Report · Anonymous (Referee 2) · 2022-4-28

Strengths

This paper suggests a novel path to study an analog of moire flat bands with cold atoms, specifically for the scenario of quadratic band touching. The theoretical analysis is very thorough, and the identify key features that reproduce qualitative aspects sought after in twisted bilayer systems (e.g., the introduction of flat and gapped bands) as well as other phenomena (quasiperiodicity induced localization transition) which may be more specific to the suggested cold atom implementation. I find it to be a major strength that the suggested approach captures the essential features / spirit of twisted bilayer systems (induced flat & gapped bands) without attempting to jump through hoops to actually implement a bilayer system with twisting (which is rather impractical based on traditional cold-atom techniques).

Weaknesses

I have not identified any serious weaknesses. While I ask for some added input in my report below, this does not signify any weakness of the work.

Report

I believe that this paper, which I find to be original, timely, thorough, and thought-provoking, meets the acceptance criteria of Scipost. While there have been previous theory papers suggesting to study the physics of twisted bilayer systems with cold atoms (and one experimental realization), this paper sets itself apart both by focusing on the case of quadratic band touching points as well as by suggesting a path to explore the essential physics of moire-induced flat bands without actually going through the rather unnatural path of implementing bilayer/multi-layer twisting with cold atoms.

Given all of the above points, I suggest acceptance of this paper.

Requested changes

  1. a bit more description on the suggested optical lattice implementation

Similar to Referee 1, I think that the paper would benefit from a bit more guidance to experimentalists who wish to follow up on this nice work and perform this kind of implementation with cold atoms. Looking at Ref. 36, which first discusses implementations of such quadratic band touching Hamiltonians with cold atoms, I find it to bit opaque (at least for an experimentalist). It would be helpful if the authors could specify, perhaps in an appendix, the form for the physical potential that they consider, and more specifically the relevant parameter values or ratios (say, the values of V_1 and V_2 from Eq. 1 of Ref. 36) that one should implement to achieve the desired form of their assumed tight-binding model.

The incommensurate lattice discussion is quite explicit, and so my only request is a bit more explicit discussion on the base lattice potential providing the discussed tight-binding model.

  1. On page 10, the authors state "As a result, this magic-angle transition results in a complete destabilization of the QBT phase." This is in the context of discussing the behavior at high W, where essentially one finds the onset of the canonical Aubry-Andre(') transition, leading to momentum delocalization (real-space localization). It is my suspicion that this behavior -- the onset of a quasiperiodicity-induced localization transition, is specific to the suggested implementation of moire physics, and may not be relevant (or at least not as directly relevant) to the analogous case of actual twisted bilayer/multilayer systems. I would have two requests -- (a) that the authors address this (whether it's specific to their proposed model, or if they suspect it to be more general and potentially relevant in real twistronic systems) and (b) that the authors change this specific sentence to read something more like "...this moire transition..." (because it's not physically a magic angle scenario), if in fact this kind of localization transition is not expected in the actual twistronic systems.

  2. In the first paragraph of the Discussion (section V), the authors discuss issues relevant to the implementation with cold atoms. It may be relevant to address that the equilibrium exploration of this physics requires studying atomic Fermi gases at rather high densities / fillings (high chemical potential), so as to populate the higher bands. There has not been much work on atomic Fermi gases in this regime to date, but theoretical proposals of this sort present a nice motivation for researchers to push their systems to achieve this capability.

  • validity: high
  • significance: high
  • originality: high
  • clarity: high
  • formatting: good
  • grammar: excellent

Author:  Junhyun Lee  on 2022-05-24  [id 2510]

(in reply to Report 2 on 2022-04-28)

  1. The most important constraint on the values of $V_1$ and $V_2$ is $V_2 > V_1 / 2$. This is to ensure the optical lattice has two energy minima at the bond centers, and not a single minima at the unit cell center (please see Supplementary information S1 of Ref. 36 for figures). Within this constraint, additional tuning of $V_1$, $V_2$ changes the depth of the minima (compared to the unit cell center point) that will affect the tight binding parameters. Since the parameters tuned in the experiments will be $V_1, V_2$ and not the tight binding parameters, we used the same method in Supplementary information S2 of Ref. 36 to find the parameters $V_1, V_2$ that gives a nice QBT. $V_1 = 2.4E_R$ and $V_2 = 1.6E_R$ which band structure is depicted in Fig. S2 (c) of Ref. 36 is indeed a good choice and are the suggested parameter in the reference. Concentrating just at the QBT, $V_1 = 1.6E_R$ and $V_2 = 1.2E_R$ results in a slightly more isotropic and symmetric (between the upper and lower band) QBT. We believe any pair of $(V_1, V_2)$ interpolating the two suggestions will work as well. One note is that in the experimental suggestion, the filling should be 3 and not 2. This is because the symmetry between the two QBT's are only present after the approximation to the tight-binding model, and the low-energy QBT in the tight-binding model cannot be realized in experiment within reasonable parameters. However, all the conclusions in our paper is from the symmetric tight-binding model, and thus applicable to the high-energy QBT as well, which is the QBT at filling 3. Together with the information on point 1 of the first referee, we made a new section Appendix A, and included the answers.

  2. We agree with the referee's comment that the destabilization of the QBT is essentially the physics one observes in high $W$ Aubry-Andre models, though its occuring in a higher dimension allows for critical phases to appear. And with the analogy between the twistronic systems and quasiperiodic systems, we also believe that the analogous phenomena can happen in the bilayer/multilayer systems. However, the issue with multilayer systems are the limited controllability, especially on $W$. The high $W$ of this transition is much larger than the experimentally accessible value of $W$, unless some breakthrough method comes through and enables to tune $W$ orders of magnitudes. It is thus in principle relevant, but realistically irrelevant in this sense. We change the sentence as suggested by the referee, and add another sentence explaining on what sense we think this is irrelevant for current twistronic systems and how such a transition could be observed using ultracold atoms.

  3. We agree on the referee's comment. The filling we consider in our work is the filling of 2, and in experimental realizations the filling should be 3. (Note the three band model we consider is above the $s$-channel, which corresponds to filling of 1) As in Fig. 2, there are two QBT's and we have considered the one with lower energy at the $M$-point (thus, filling of 2), but experimentally the higher energy QBT at the $\Gamma$-point is more relevant which is at the filling of 3. We have addressed in the response to comment 1, that theoretically the two QBT's are equivalent in the tight-binding limit. To achieve a high chemical potential for the atomic Fermi gases to have a stable filling of 3 is required to realize this physics. We address this point in the first paragraph of Section V, in the hope of motivating experimental effort.

---

## Round 2 · Referee Report · Anonymous (Referee 2) · 2022-5-24

Report

The author's have addressed all of my questions and concerns, in a very satisfactory manner. I continue to recommend publication.

---

## Round 2 · Referee Report · Anonymous (Referee 1) · 2022-6-9

Report

The authors have satisfied my concerns and questions from the last round of review through their detailed response as well as clarifications in the manuscript. I would also like to clarify the confusion I left in the 4-th question of my previous review. There, I was intended to ask how the solid line in Fig.3 will move upon increasing the order of the perturbation, which is merely from the perspective of presentation of this result, and I encourage the authors to also include the higher-order result in Fig.3. Nevertheless, the point that the higher-order perturbation prediction will move towards the numerical result is clear. In this sense, I recommend the manuscript to be published.

---

## Round 2 · Author Response

We would like to thank both referees for the detailed reading of our manuscript, and for important and constructive suggestions. We have now modified the manuscript accordingly, and we post the detailed response to the comments below as a reply.

---

## Round 2 · List of Changes

1. We added Appendix A: Experimental realization, which explains the experimental parameters for the optical lattice and pre-tight binding approximation band structures. We also added a sentence in Sec. II referring to this appendix.

  2. In Sec. IV, we clarified that the transition at large $W$ is a moir\'e transition and how this is related to the magic-angle transitions.

  3. In Sec. V, we emphasized that high chemical potential is needed for the experimental realization.

  4. We specified that the interaction can be tuned via a Feshbach resonance in Sec. V and included the references.

All changes are colored in blue in the resubmitted manuscript.

---

## Editorial Decision

published